# A Mixed-Methods Assessment of Residential Housing Tenants’ Concerns about Property Habitability and the Implementation of Habitability Laws in Southern Nevada

**DOI:** 10.3390/ijerph19148537

**Published:** 2022-07-13

**Authors:** Erika Marquez, Courtney Coughenour, Maxim Gakh, Tiana Tu, Pashtana Usufzy, Shawn Gerstenberger

**Affiliations:** School of Public Health, University of Nevada, Las Vegas, Las Vegas, NV 89154, USA; courtney.coughenour@unlv.edu (C.C.); maxim.gakh@unlv.edu (M.G.); tut1@unlv.nevada.edu (T.T.); usufzyp@unlv.nevada.edu (P.U.); shawn.gerstenberger@unlv.edu (S.G.)

**Keywords:** substandard housing, housing quality, housing policy, habitability, healthy housing

## Abstract

Housing is a key health determinant. Habitability laws set minimum standards for adequate housing. However, accessing them to ensure adequate housing may be a challenge for many tenants. This paper explores the need for rental housing policy that would better support adequate and safe housing, particularly for low-income renters. A mixed-methods approach assessed residential tenant habitability concerns in Clark County, Nevada, through calls relayed to the Clark County Landlord–Tenant Hotline (CCLTH). Of the 2865 calls, 74.3% were from ZIP codes that were 80% of the median income and below. There was a significant relationship between the ZIP code-level income and the reporting of at least one essential habitability concern. Of the 266 participants that responded to a follow-up call, 34.6% reported that their complaint was resolved and there was no association between resolution and income. Qualitative data analysis from phone interviews revealed two central themes: (1) resources to navigate landlord–tenant laws are limiting and (2) housing policies need to be strengthened to help tenants and keep people housed. Understanding tenant concerns regarding substandard housing and related inequities can help inform rental housing policy and its implementation to promote healthy homes and improve health outcomes for communities burdened by poor rental housing conditions.

## 1. Introduction

Housing is fundamentally tied to human health. It is widely acknowledged as a social determinant of health, being one of several key social factors that shape health behaviors and outcomes [1,2]. Households experiencing inadequacies in housing conditions can have negative physical health outcomes [3]. Substandard housing conditions, including improper waste disposal, lack of hot water, pest infestations, and crowding have been linked to the spread of infectious diseases [4]. Similarly, a range of chronic diseases appear connected to such hazards. Lead, for instance, can be found in housing in the form of leaded paints and plumbing and has been linked to “respiratory, neurologic, digestive, cardiovascular and urinary diseases” [5]. Furthermore, dampness, mold, and pest infestations have been found to exacerbate allergies and asthma, and exposure to radon or asbestos is associated with the development of lung cancer [6,7,8].

The mental health of individuals residing in substandard housing suffers as well. One systematic review of nine studies examined the effect of “living in cold and damp homes” and concluded that it negatively impacts both mental well-being and mental disorders [9]. Housing quality, alongside a household’s ability to deal with financial problems, has been shown to have a small but significant effect on mental health [10]. A 2019 systematic review of 12 longitudinal studies confirmed associations between housing disadvantage and stress, depression, anxiety, mental strain, and worse mental health scores [11].

The relationship between housing quality and health also reveals inequities: those of lower socioeconomic status and communities of color bear a disproportionate burden. The 2019 American Housing Survey reported that 9% of households with an annual income below USD 20,000 reside in housing classified as inadequate, compared to less than 3% of those earning USD 60,000 or more [12]. Furthermore, African American and Hispanic households were twice as likely as white households to live in housing that was “severely inadequate,” defined as housing with a lack of complete kitchen, plumbing facilities, overcrowding, and/or severely cost-burdened occupants [12]. Although there is limited research on the dynamics between landlords and tenants, there are instances where landlords simply do not attempt to remediate hazards—taking advantage of vulnerable households by neglecting property maintenance and being unwilling to allow tenants to make changes [13]. The combination of unsatisfactory landlord–tenant dynamics concerning housing affordability and poor housing quality and safety puts renters, particularly those of low-income and/or racial and ethnic minority groups, at risk. One of these risks is the possibility of involuntary displacement in particularly unstable housing situations. Involuntary displacement takes physical, emotional, and financial tolls on families [14].

Addressing housing hazards can help improve the health outcomes of renters, especially those from low-income and racial minority groups. This is especially important in Clark County, Nevada. The county has a high percentage of rental housing units, with 45.2% of housing units being renter occupied [15]. In Las Vegas alone, 14.4% of rental units are moderately inadequate and 5.6% are severely inadequate, with physical problems that range from plumbing, heating, electric, wiring, or upkeep concerns [16]. Data from the 2020 ACS five-year estimates suggest that out of 809,026 occupied housing units in Clark County, almost 90,000 completely lack plumbing or a kitchen [15]. Efforts that utilize rental housing policy to support tenants have been shown to positively impact health outcomes [17].

Chapter 118A of the Nevada Revised Statutes (NRS) outlines the baseline habitability obligations and rights of landlords and tenants, as well as legal remedies in the case of disputes. However, despite the clear link between health and housing, the local health department currently has no authority to enforce these obligations. Instead, the most readily available path for many Clark County tenants trying to exercise the habitability rights established by state law is to turn to the judicial system. Tenants also have the option to call the Clark County Landlord–Tenant Hotline (CCLTH) when confronted with housing habitability issues. The CCLTH is managed by the local health department and university and provides verbal guidance on the proper, lawful steps to report housing-related issues to landlords. From 2011 to 2013, CCLTH documented 3523 total housing complaints [18]. Among the most common complaints were indications of mold, bed bugs, the necessity for general maintenance, and cockroaches in the residence [18]. In addition, complaints received by the CCLTH suggest that existing policies are not meeting tenants’ needs.

The overall goal of this mixed-methods assessment was to examine the need for housing policies that support quality housing for low-income renters in Clark County, Nevada. We aimed to examine the problems with rental housing stock in properties occupied by tenants who felt aggrieved enough to make a complaint to the CCLTH and to examine the tenant perspectives regarding policies that could support quality housing. We used quantitative data to examine habitability concerns from tenants who contacted the CCLTH to describe the types of complaints and to assess the relationship between essential service complaints and ZIP code median household income to determine if ZIP code median household income is associated with the likelihood of having habitability concerns and of having the concerns resolved. We used qualitative data to understand tenants’ perceptions of the existing enforcement and implementation, including the utility of the CCLTH to help address housing concerns and views on the need for supportive rental housing policies that would expand the role of the local health department in enforcement.

## 2. Materials and Methods

This assessment used a mixed-methods study approach. Mixed-methods research combines qualitative and quantitative data analysis in one study. Integrating qualitative and quantitative data can cultivate a deeper understanding of results when exploring complex research topics [19]. Data were collected from tenants who called the Clark County Landlord–Tenant Hotline (CCLTH), and from a smaller subset who participated in semistructured interviews to examine the lived experiences of tenants living in rental housing. This study was approved by the University of Nevada, Las Vegas (UNLV) Institutional Review Board (IRB) number 760803-7.

### 2.1. Clark County Landlord–Tenant Hotline

The CCLTH was developed through a partnership between the University of Nevada, Las Vegas School of Public Health and the local health department, the Southern Nevada Health District (SNHD). The hotline’s focus is on documenting the prevalence and type of habitability concerns reported by renters in Clark County, Nevada. When confronted with housing habitability issues, Clark County tenants may contact the CCLTH for verbal guidance on the proper, lawful steps to report housing-related issues to landlords. Habitability concerns of callers are categorized into essential and nonessential services. Essential service complaints include the sporadic or permanent loss of HVAC services, water, electricity, or gas. Nonessential services include complaints such as mold, roaches, rodents, general maintenance (e.g., appliances, electrical issues, fire damage, etc.), bedbugs and other insects (all bugs other than roaches and bedbugs), odor, sewage, domestic animal, pigeon, hoarder, or other.

Similar to other jurisdictions, state landlord–tenant statutes exist to set baseline habitability standards and provide conflict resolution mechanisms for landlords and tenants. However, tenants in Clark County must either be extremely knowledgeable about these legal processes to exercise their rights as tenants effectively or have access to attorneys. Therefore, in addition to capturing the prevalence of housing concerns, the CCLTH served as a resource to help guide tenants on how to report housing concerns to their landlords and provided information on legal options.

### 2.2. Nevada Housing Statutes

In Clark County, Nevada, rental housing is governed by leases and also by the landlord–tenant Chapter of the Nevada Revised Statute (NRS) [20]. In relevant parts, the NRS requires landlords to maintain a “dwelling unit in a habitable condition” [21]. Habitability is generally a function of access to certain essential conditions within the dwelling, including waterproofing, plumbing, heating, electricity, sanitation, and maintenance of basic structures such as walls, floors, and ceilings [21]. Additionally, per NRS § 118A.290, landlords must “maintain the dwelling unit in a habitable condition. A dwelling unit is not habitable if it violates provisions of housing or health codes concerning the health, safety, sanitation, or fitness for habitation of the dwelling unit” [21].

The NRS also outlines a process for tenants who inhabit dwellings that lack an “essential item” [21]. After receiving a written complaint from a tenant alleging that the unit is not habitable due to a lack of “essential items or services” that are required by a lease or the NRS (e.g., lack of heating, air conditioning, electric, gas, or water), and that the “landlord willfully or negligently fails to” provide, the landlord has 48 h to “remedy the breach, or use his or her best effort to remedy the breach” [21]. Failure of the landlord to do so can allow the tenant to recover certain damages in a legal action, withhold rent during noncompliance if rent is otherwise current, and procure comparable housing during noncompliance [21]. Alternatively, if the tenant is not responsible for causing the uninhabitability, and in instances where the cost of repair is relatively low, the tenant may also notify the landlord of an intention to remedy the problem at the landlord’s expense, and after a period of time, complete the maintenance [21]. Despite these requirements, a landlord’s financial liability for failing to provide habitable conditions is limited [21].

### 2.3. Study Design

A cross-sectional, retrospective study design was used to capture tenant habitability concerns and housing dispute resolutions from tenants who called the CCLTH. The qualitative portion of the study used a phenomenology study design to explore a particular phenomenon within a group of people who have all experienced a similar event [22], defined here as being a renter in Clark County whose housing conditions precipitated a call to the CCLTH between May 2011 and April 2013. The inclusion criteria consisted of callers who provided a ZIP code within Clark County, had a documented rental housing complaint, and did not identify as an owner of a housing unit.

#### 2.3.1. Quantitative Data

We used quantitative data to describe the types of complaints made by tenants to the CCLTH between 2011 and 2013 to assess the relationship between essential service complaints and the ZIP code median household income. We aimed to determine if ZIP code median household income is associated with the likelihood of having habitability concerns and having the concerns resolved. Tenant habitability concerns and residential ZIP codes were extracted from conversations with callers who contacted the CCLTH. Median income per the 2015 U.S. Census five-year estimates of the ZIP code in which the residence was located was recorded. Tenant habitability concerns were categorized as essential or nonessential (based upon the above criteria for essential and nonessential services). Callers received a follow-up phone call between March 2014 and May 2015 to assess if their housing concerns had been resolved. Two attempts were made to reach tenants. If CCLTH staff reached the original tenant who expressed habitability concerns, the tenant was asked to self-report if the initial housing concern had been corrected.

#### 2.3.2. Qualitative Data

Seventeen in-depth, semistructured interviews were conducted with a subset of Clark County tenants who contacted the CCLTH between June and August 2015 about their experiences living in rental housing. Participants were selected using a purposeful sampling technique. These data were collected as part of a larger qualitative study. In this analysis, we focus on interview data that were collected to better understand the utilization of the CCLTH, the use of legal resources to address housing concerns, and tenants’ perceptions about the need for supportive rental housing policies that would expand the role of the local health department in enforcement (Appendix A).

### 2.4. Data Analysis

#### 2.4.1. Quantitative

A Pearson’s chi-squared analysis examined the association between tenant habitability concerns and income. First, CCLTH callers with at least one essential service complaint were identified. Second, callers were grouped into categories based on ZIP code level as being either above or below median income—as defined by the 2015 U.S. Department of Housing and Urban Development’s (HUD) guidelines for 80% of the metropolitan area’s median household income for a family of four—where less than 80% of median income was considered low income. A Pearson’s chi-squared test determined the relationship between household income and the lack of essential services, with a significance of α = 0.05.

Similarly, a contingency table was used to dichotomize income into below and above 80% median income and whether the housing deficiency was corrected to assess if tenant income was associated with the likelihood of having a tenant habitability concern resolved. A housing deficiency was considered corrected if the initial tenant habitability concerns were resolved per self-report by the tenant during the follow-up call. A Pearson’s chi-squared test determined the relationship between ZIP code median income and having housing deficiencies corrected with a significance of α = 0.05.

#### 2.4.2. Qualitative

The research team used an inductive approach to analyze interview data. This approach relies on specific objectives to derive concepts and themes from the collected data [23]. All phone interviews were transcribed and coded using thematic text analysis—a long-proven methodology in qualitative research. Thematic text analysis established codes (topics and subtopics) that were useful in identifying central themes from the lived experiences of participants [24]. To ensure reliability of qualitative data analysis, the coding process followed the intercoder reliability and agreement process. Intercoder reliability involves two or more equally capable coders independently analyzing a set of qualitative data; the agreement process requires the coders to reconcile coding discrepancies through discussion [25]. This methodology is similar to that described by Marquez, Dodge Francis, and Gerstenberger [14].

## 3. Results

### 3.1. Quantitative

#### 3.1.1. Summary and Assessment of Tenant Habitability Concerns by Income

A total of 3713 calls were received by the CCLTH during the study period, of which 2865 calls met the inclusion criteria. Calls were broken down as the ZIP code median income being either above or below 80% of the Clark County median household income—estimated to be USD 59,200 from the 2015 ACS five-year estimates [26]. These two groups were further subdivided into ZIP codes with a median income below 50% of the median household income, between 50% and 80% of the median household income, between 80% and 100% of the median household income, and above the median household income. The highest proportion of calls were from ZIP codes that were 80% of median income and below (lower-income community), comprising 74.3% of total calls received. Tenant habitability concerns were categorized into essential and nonessential. Heating, ventilation, and air conditioning (HVAC) outages, water outages, and electric or gas outages were categorized under essential tenant habitability concerns. Most of the reported concerns in this category were regarding HVAC outages, comprising 13.7% of the total calls. Other hazards, such as mold, bed bugs, sewage, and other insects/animal infestations, were categorized as nonessential habitability concerns. The majority of nonessential calls were for mold (34%) and general maintenance (32.5%). Table 1 summarizes all calls received by ZIP code income and complaint category.

The chi-squared analysis indicated a significant relationship between income and reporting at least one essential habitability concern (X^2^ (1, N = 2865) = 5.566, *p* < 0.05). Nearly three-quarters of calls made to the CCLTH involved tenant habitability concerns from ZIP codes below the income threshold. Thus, the CCLTH received a disproportionate number of tenant habitability concerns from lower-income tenants compared to those who earned above 80% of median income. This is further illustrated in Figure 1, which depicts housing complaints by median income for each ZIP code. The data indicate that lower-income ZIP codes are particularly vulnerable to tenant habitability concerns.

#### 3.1.2. Addressing Housing Concern Resolutions by Income

A total of 266 tenants responded to follow-up calls in which they were asked to self-report if their housing concerns had been resolved. Of those who responded, 34.6% self-reported having their complaint resolved, of whom 67.4% were at or below 80% median income and 33.3% were above 80% median income. See Table 2 for resolutions by ZIP code median income level. Chi-squared analysis indicated no significant association between ZIP code income and a resolution (X^2^ (1, N = 266) 0.095, *p* = 0.76).

### 3.2. Qualitative

The qualitative data collected from participant interviews provided important insight into how renters navigate existing housing policies and resources and explored what tenants think would improve habitability concerns among rental units. The transcriptions of the interviews were followed with thematic text analysis, wherein two central themes arose: (1) resources to navigate landlord–tenant laws are limited and (2) housing policies need to be strengthened to help tenants and keep people housed.

#### 3.2.1. Theme: Resources to Navigate Landlord–Tenant Laws Are Limited

A limited number of resources are available for tenants to navigate landlord–tenant laws. These laws may serve an essential function in aiding tenants who are able to navigate the system or to upkeep housing. However, most interviewees found the Clark County Landlord–Tenant Hotline (CCLTH) a helpful resource to understand their rights and how to report tenant habitability concerns. One tenant described what precipitated a call to CCLTH:

“Like the only thing I kept doing was going in there … complaining to her, like what’s going on with this apartment? … So definitely, when I called the hotline … they went [through the process] what I’m going to do, how to go about doing it.”(Interviewee, 003)

Although the CCLTH was an initial point of contact and was found to be helpful, most participants acknowledged its limitations. One tenant indicated:

“Um, somewhat but they really couldn’t do anything for the situation. Uhm, they just like told me kind of how to go about it.”(Interviewee, 012)

It was apparent through the interviews that the limitations of the CCLTH were largely because the local health department lacked enforcement authority. One interviewee commented about the local health departments authority:

“They tried but their hands are tied. They can’t. The program can’t do anything but come and look and put down on paper. They can’t stop these people. They can’t assist you with anything else. So you know as far as being helpful. Yes they came out and looked [but] they couldn’t help.”(Interviewee, 002)

Several participants suggested ways in which the hotline could be improved to service the needs of renters in Clark County. Suggestions included advocating on behalf of tenants rather than dispensing information alone and enforcing housing policies to hold landlords accountable. Take, for example, this participant’s appeal for more enforcement authority:

“The only problem is the when it pertains to my situation with mold there is no strict regulation on how they repair it … the issue is there is no regulation saying the landlord has to get this certified mold removed by a mold tester. Before he can say hey it’s done its clean and put the wall back together. He didn’t even want to tear the wall apart.”(Interviewee, 004)

Another alternative for Clark County residents is to seek assistance through the court system by filing a civil complaint against their landlord. The Legal Aid Center of Southern Nevada and Nevada Legal Services are resources to support lower-income tenants through the process. Of those who participated in the study and contacted one of these agencies, there was no unanimous consensus on how effective these experiences were in helping navigate the system. Among the participants who found these services to be helpful, they stated that it aided them in dealing with situations such as serving paperwork and five-day notices. One interviewee described Nevada Legal Aid Center’s role in helping to address housing remediation:

“… when I got my 5-day notice, I went down there and they helped me do the paperwork myself so that I could go to court … they helped out a whole lot as far as me having to serve [my landlord] with papers so that he could fix stuff …”(Interviewee, 013)

There were other participants who found these legal assistance resources to be frustrating and unhelpful in resolving their cases in ways that they perceived to be appropriate and just. One individual describes dissatisfaction with attempting to resolve a case concerning heat treatments using one of these resources:

“I went and asked for help but … they were not helpful either. They said it was like a 50/50 chance and … since the landlord he went and did a heat treatment … if he did all that then we were probably going to lose the case and then we were going to end up paying the landlord for like the lawyer and stuff … We went through losing all our stuff…”(Interviewee, 005)

Most interviewees perceived varying degrees of helpfulness with the available resources in addressing tenant habitability concerns in ways that they thought were appropriate and just, including the Clark County Landlord–Tenant Hotline and legal services.

#### 3.2.2. Theme: Housing Policies Need to Be Strengthened to Help Tenants and Keep People Housed

A series of questions were asked to understand renter perceptions about housing policies that would give the local health department enforcement authority over rental housing. Specifically, these questions focused on how interviewees felt about the policy to establish this authority and the potential benefits or disadvantages to adopting and implementing it. What distinctly emerged during the interviews was the need to change current policy and tenant recommendations to improve landlord accountability or enforcement by the government.

Many participants were clearly frustrated with their housing situations. All participants wanted a safe and healthy place to live. They expressed wanting accountability among “slum” landlords. Of those who provided feedback regarding the possible adoption and implementation of the rental housing policy, no one expressed concerns about the proposed policy, and all indicated some degree of need.

Tenants also expressed several recommendations to improve current rental housing conditions that could be used to guide policy. A common recurring construct articulated by participants involved using third-party liaisons to help enforce landlord–tenant policies. Some indicated that a qualified third-party inspection process should take place before market listing to ensure that hazards were addressed and remediated. After an initial inspection prior to market listing, participants articulated that it was also important that the standard of housing be upheld. This tenant describes:

“Well like when it’s a situation where the apartment is possibly uninhabitable due to either health risk or even if it’s something to do with the landlord neglecting, uhm, you know repairs … We need to have some sort of third party, you know? The health district, or whomever, would be part of that to say, ‘hey you know this dwelling is not inhabitable for your tenant. You know if you don’t fix this within a sort of amount of time we are going to allow your tenant to either hire someone or they’re going to get out of the lease.’”(Interviewee, 004)

Especially during housing inspection and repair processes, many participants explained that they experienced housing displacement. They suggested that policies should ensure providing an “equivalent” rental in such cases. One tenant offered a recommendation:

“They should find a place or they should find another unit that’s similar to ours and the same rent that they pay for or just give it to us for free until we can move back into our unit and pay that next month’s rent.”(Interviewee, 006)

As mentioned previously, participants also shared instances of bad business practices, wherein landlords did not address habitability concerns appropriately. Some participants recommended that, to encourage accountability of landlords, those who violate housing statutes should be fined. Consider this suggestion that landlords should incur a fee costlier than the expense of the corresponding hazard remediation:

“… I think that would be better like if they were able to be fined or something was to happen to where it would cost them more money to pay what or to do whatever than to pay to fix the unit. I think that would make it so much better.”(Interviewee, 006)

Many participants also emphasized the need for more transparency in landlord–tenant relationships and in how landlords conduct business. One tenant describes the idea of stating all policies to prospective renters, especially those on housing accommodations:

“… the landlords have to like specifically state all policies … As well as giving the tenants more options for emergency cases or difficult cases. Like, it’s one thing if I just said oh I don’t like this apartment I want to move. It’s another for me to tell you that I don’t feel safe the fact that you have to send a letter to tell me it’s not safe … and you not be able to accommodate that in a better manner.”(Interviewee, 014)

Another caller indicated that prior disclosures regarding habitability concerns should be established as well:

“Guidelines [that] you are supposed disclose any prior leaks. Major leaks. Any issues with the unit. And they don’t do that. And if they don’t and you find out. They should face a fine. Because that is the law. They are supposed to … disclose anything that was prior wrong with the unit.”(Interviewee, 002)

Through interviews with the participants, it was made clear that they envision specific policy recommendations that enhance transparency and protect the health of tenants.

## 4. Discussion

Because housing conditions are critical drivers of health, tenants in substandard housing are at increased risks for worse health outcomes. Compared to homeownership, renting has been correlated to disproportionate burdens and health outcomes [27]. To address such disparities, this study attempted to identify common rental housing habitability complaints and related inequities that can be used to inform rental housing policy in Clark County, Nevada, with a focus on supporting quality housing for low-income renters. We also explored perceptions about the existing enforcement and implementation of policies aimed at habitability, including the utility of the CCLTH to help address housing concerns and views on the need for supportive rental housing policies that would expand the role of the local health department in enforcement. 

There is a significant inequity present in low-income communities associated with rental housing conditions. This study found that tenants living in ZIP codes at or below 80% of the estimated median household income were more likely to call the CCLTH with habitability concerns compared to those who lived in ZIP codes above 80% of the estimated median household income. Nearly three-quarters of all the calls made to the CCLTH involved tenant habitability concerns from ZIP codes below the income threshold. However, ZIP code level income was not associated with whether or not tenants self-reported a resolution concerning the habitability issues. By understanding tenant-reported habitability concerns and related inequities, policies can be tailored to improve outcomes for communities burdened by poor rental housing conditions, particularly for vulnerable tenants who are low-income.

Similar to national and state-level statistics, we found that tenants who called the CCLTH with habitability issues were more likely to reside in low-income ZIP codes. It may also be the case that more rental housing exists in lower-income ZIP codes. Nevertheless, this is particularly concerning, as individuals residing in low-income ZIP codes are overburdened with other related factors associated with poor health outcomes, such as neighborhood crime and violence [28], inadequate physical activity amenities [29,30], and poor food environments [31]. Furthermore, families of color are also over-represented in low-income neighborhoods and are more likely to be extremely low-income renters compared to white households [32,33]. As of 2019, 21.4% of Clark County’s residents were born outside of the U.S., which is higher than the national average of 13.7% [34]. There is research that has correlated the lack of filing tenant habitability concerns with low-income status due to worries regarding retaliation in the form of eviction, rent hikes, and intimidation [35]. In addition, immigrant populations with housing deficiencies are less likely to submit tenant habitability complaints out of fear of eviction or deportation [13]. These fears may apply to involvement from a local health department as well. Given the large number of immigrants in Clark County, these data show the importance in improving housing conditions among immigrant populations.

The compounding of multiple determinants of health can concentrate disadvantage, enhancing disparities in poor health. We did not find an association between ZIP code level household income and self-report of resolution of the habitability complaints. From the standpoint of resolving disparities in housing conditions and their related health disparities, this was a positive finding, given that mitigation of hazards is likely to attenuate negative health consequences. Our finding that low-income renters are equally as likely to achieve mitigation suggests no major income-related disparities in remediation. On the other hand, only about one-third of all tenants, regardless of income, self-reported that their deficiency was mitigated. Therefore, an alternative explanation is that so few mitigations are achieved that the association between ZIP code level household income and self-reported resolution is difficult to detect. Furthermore, although there may not be an income-based disparity, the overall low rate of remediation suggests an absolute problem that should be addressed across tenant groups.

State-level habitability laws differ across the United States but share some common elements [36]. Willis and colleagues encouraged research to better understand how habitability laws are enforced and how enforcement impacts the utility of habitability laws [36]. Determining whether Nevada’s habitability laws “work” requires asking how they impact tenants, especially those who are most vulnerable. This study suggests that Nevada’s habitability laws—and how they are implemented in practice—leaves many tenants with difficulty navigating their requirements, and potentially unprotected from habitability issues. An alternative explanation is that many other tenants live in habitable housing and maintain their access to essential services and therefore have no reason to call the hotline. However, this study demonstrates that many low-income tenants, and likely tenants who are ethnic minorities and who live in poverty, continue to struggle with maintaining habitable housing, which includes essential services. Hotline callers may represent tenants with some of the most drastic housing deficiencies in Clark County. However, those who called the hotline had the knowledge, time, confidence, and ability to do so. There may be even more tenants who live in poor housing conditions but cannot or do not call the hotline, suggesting that estimates of inhabitability from this study may undercount the problem. Additional research is needed to understand more comprehensively how many residential rental properties are uninhabitable or lack essential services and the disparities involved. Such research may shed light on whether Nevada’s habitability laws “work” to protect the most vulnerable tenants.

This study supports the need for stronger policies to ensure access to habitable housing with access to essential services as a public health strategy. Willis and colleagues [36] recently ranked Nevada’s habitability laws as some of the strongest in the U.S. Even so, enforcement of the existing laws may be lacking. If habitability statutes are a factor driving the minimum quality standards for rental housing, then our findings are especially concerning in light of this ranking; they may suggest that tenants in other states are faring even worse.

Evaluating the impacts of habitability laws comprehensively also requires examining their implementation and enforcement, as Willis and colleagues [36] suggest. Increasing access to resources such as the CCLTH, the local health department, and legal aid services may lend some help. However, as the participants suggested, relying on these entities in their current forms may also lead to frustration because of their limited authorities and capacities. Pursuing legal remedies, with legal assistance or independently, may frustrate tenants for yet another reason—it may unearth the limitations of the judicial system to produce outcomes that participants view as fair and just. Nevertheless, with alternative avenues to resolve landlord–tenant habitability disputes being absent, participants are left to turn to the judicial system. Access to attorneys may help low-income tenants navigate the currently complex processes. This involves providing additional resources to legal aid organizations. The Legal Services Corporation estimates that 62–72% of low-income individuals who sought legal aid funded by the Legal Services Corporation nationwide received either no legal help or inadequate legal help due to insufficient resources [37].

The findings from the interviews in this study indicate that participants may need help understanding, navigating, and exercising their habitability rights. Enabling the local health department to play a more fundamental role in enforcing habitability laws may be an opportunity to strengthen the enforcement of habitability laws. The participants clearly saw this as an opportunity. This may be an especially fruitful opportunity, given the participants’ frustration with the existing system and the possibility that a local health department with regulatory authority could alleviate the enforcement burdens currently borne by tenants—who are often vulnerable—and legal aid organizations, which are under-resourced. Regulatory roles and authorities of the local health department could be delineated to prioritize tenants who are in greatest need, housing that is in the worst shape, and landlords who may be failing to meet their duties across multiple properties or over prolonged periods of time. For instance, regulatory roles can involve granting authority to local health departments to establish an inspection system for rental units. Enforcement for inspection should set requirements, such as specific safety and remediation standards, fees, timely obligations, and other conditions, to ensure quality housing. Furthermore, support for quality rental housing does not have to be limited to local health departments only. Funding nonprofit organizations can also expand the system to support quality rental housing.

Although we were able to better understand habitability complaints and perceptions of navigating the current systems, our study does have some limitations. The CCLTH does not collect demographic data, making inferences about the population difficult. Because household incomes for participants were not collected, estimates were therefore based on ZIP codes. Individual-level data would have enabled a more granular analysis. The complaint data captured from the CCLTH were based on a convenience sample and are subject to bias of those who actually contacted the hotline compared to the entire population that may have needed the service. Because housing complaints were self-reported by callers, it was difficult to ensure the validity of concerns. It was also difficult to ensure how well participants understood the current and proposed rental housing policies. Some participants were unavailable for follow-up calls as well, which caused a loss of some data and may not truly represent perceptions on remediation from all who initially called the CCLTH during the study period. It should be pointed out that interviews did not include landlords to gain their perspectives. Although we completed an adequate number of interviews to attain saturation [38], the captured data still may not represent the population of renters in Clark County.

## 5. Conclusions

In urban centers such as Clark County, Nevada, which are populated by diverse, vulnerable populations, ensuring stable and quality housing is critical. Housing is a fundamental determinant of health, and housing policies are at the forefront to support quality housing—specifically for low-income renters. More work needs to be conducted to understand how the power imbalance in landlord–tenant relationships drives housing habitability concerns and displacement. However, qualitative and quantitative data from this study support the need for stronger housing policies. Without such policies, disparities and inequities will continue to persist among our most vulnerable populations.

## Figures and Tables

**Figure 1 ijerph-19-08537-f001:**
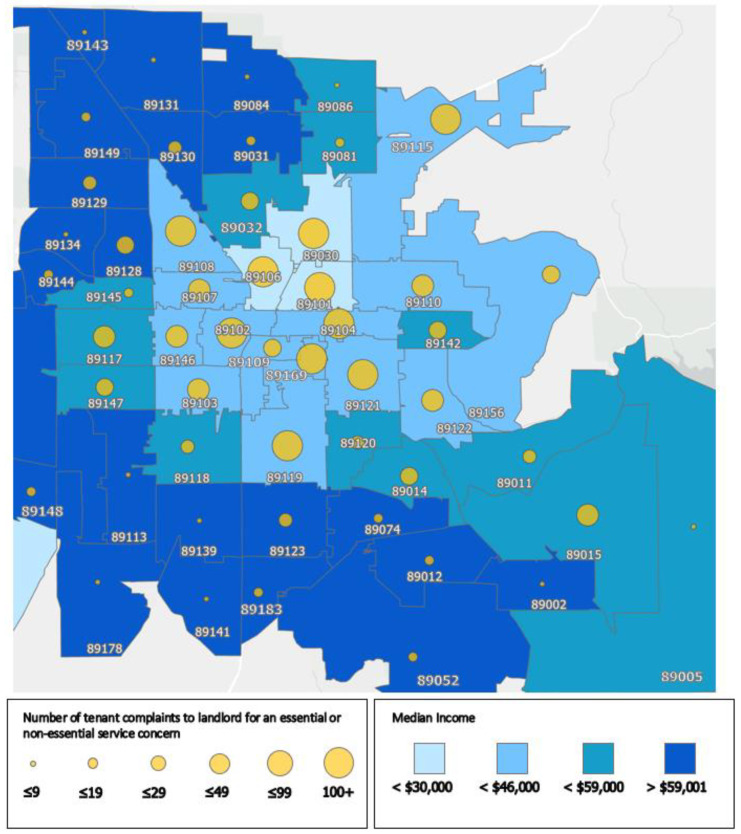
Housing habitability concerns from 2865 calls made to the Clark County Landlord–Tenant Hotline (CCLTH) between May 2011 and April 2013 by median income for each ZIP code.

**Table 1 ijerph-19-08537-t001:** Summary of complaints by ZIP code median income and essential and nonessential services from 2865 calls made to the Clark County Landlord–Tenant Hotline (CCLTH) between May 2011 and April 2013.

**Income Limits**	**N**	**%**
Very Low Income (50% of Median Household Income ^1^ ≤ USD 30,700)	459	16
Low Income (80% of Median Household Income ≤ USD 30,700–49,100)	2133	74.4
Above Median Household Income (≥USD 59,200)	273	9.5
**Essential Tenant Habitability Concerns**	**N**	**%**
Heating Ventilation and Air Conditioning (HVAC) Outage	392	13.7
Water Outage	185	6.5
Electric or Gas Outage	28	1
**Nonessential Tenant Habitability Concerns**	**N**	**%**
Mold	975	34
General Maintenance	931	32.5
Bed Bugs	540	18.8
Cockroaches	427	14.9
Other (e.g., problems with sinks, toilets, neighbors)	235	8.2
Other Insects	170	5.9
Odor	164	5.7
Sewage	131	4.6
Rodents	85	3
Pigeons	39	1.4
Domestic Animals	33	1.2

^1^ Clark County median income is USD 59,200.

**Table 2 ijerph-19-08537-t002:** Number of housing deficiencies resolved by 80% of median income.

	Deficiency Corrected	Deficiency Not Corrected	Total
At or below 80% of median income	62 (35.2%)	114 (64.8%)	176
Above 80% of median income	30 (66.7%)	60 (33.3%)	90
Total	92 (34.6%)	174 (65.4%)	266

## Data Availability

The data presented in this study are available upon request from the corresponding author. The data are not publicly available due to participant privacy.

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
