# Peer review of "A Mixed-Methods Assessment of Residential Housing Tenants’ Concerns about Property Habitability and the Implementation of Habitability Laws in Southern Nevada"

_ijerph, 2022, doi:10.3390/ijerph19148537_

Round 1

Reviewer 1 Report

The subject covered in the paper is, in my opinion, interesting and rarely taken up in the literature.

However, there are some important, in my opinion, important comments about the paper:

- Research data is quite old (complaints from tenants from 2011-2013). Why such old data? Is it possible to take into account newer data? Please explain (this issue should also be clarified in the paper). Maybe the problem described by the authors has already been solved to some extent.

- Purpose of the work - it should be clearly formulated and presented in a separate paragraph of the text (of course, you can also define specific goals or research hypotheses).

- The authors use "mixed methods". This wording even appears in the title of the paper. It is worth explaining what these "mixed methods" are and describing them.

- In my opinion, some Conclusions in the presented version are more of a summary. Mainly research conclusions should be indicated, preferably in the form of points. It's also worth adding recommendations.

Detailed Notes:

- in the first part "Introduction", subchapter 1.1 should not be separated, if subchapter 1.2 was not separated (the chapter is not divided into one subsection, in my opinion),

- in table 1, two lines are defined identically (50% of Median Household Income…) but different values. Please explain.

In my opinion, the paper can be published after corrections and explanations by the Authors.

Reviewer 2 Report

Dear authors,

Thank you for conducting a study on such an important topic that addresses inequities among minority groups to enhance housing policies. While the methods and analysis were well-stated/ written, I have minor comments related to the structure of the paper. I felt there is redundant information about the collected data. I believe that the authors may reorganize some of the information to make it concise. Perhaps the data could be mentioned at the beginning without the need to repeat it under "results". 

Second, I would suggest moving the limitations section under after the conclusions. Also, the conclusions are too short and would benefit from a summary of the proposed policies.

Third, in the results section, what were the criteria for inclusion, and are there sources to justify the inclusion criteria?

Fourth, I believe section 3.2 could be re-organized. The quotes provided could benefit from grouping them into themes or categories. Perhaps this section needs more attention on how the results are reported.

Lastly, I noticed that many of the citations are outdated with few recent ones, mostly from reports. Can the authors confirm other studies in the field that are more recent?

Reviewer 3 Report

The article is clearly written with a solid policy purpose and exposition. The quantitative section is limited but effective and the qualitative work supports the analysis well. No major issues detected.

Typo in line 2 of Table 1: should be between 50 and 80% AMI.

Round 2

Reviewer 1 Report

All my comments have been taken into account by the Authors. The authors also explained my doubts.

The paper in its current version can be published in IJERPH, in my opinion.